# Contact Hole Shrinkage: Simulation Study of Resist Flow Process and Its Application to Block Copolymers

**DOI:** 10.3390/mi15091151

**Published:** 2024-09-13

**Authors:** Sang-Kon Kim

**Affiliations:** The Faculty of Liberal Arts, Hongik University, Seoul 04066, Republic of Korea; sangkona@hongik.ac.kr

**Keywords:** computational lithography, shrinkage process, resist reflow, resist flow process, surface evolver, finite element method, deep learning, machine learning, directed self-assembly, block copolymer, self-consistent field theory

## Abstract

For vertical interconnect access (VIA) in three-dimensional (3D) structure chips, including those with high bandwidth memory (HBM), shrinking contact holes (C/Hs) using the resist flow process (RFP) represents the most promising technology for low-$k_{1}\;(where\;CD=k_{1}\lambda /NA,$
CD is the critical dimension, λ is wavelength, and NA is the numerical aperture). This method offers a way to reduce dimensions without additional complex process steps and is independent of optical technologies. However, most empirical models are heuristic methods and use linear regression to predict the critical dimension of the reflowed structure but do not account for intermediate shapes. In this research, the resist flow process (RFP) was modeled using the evolution method, the finite-element method, machine learning, and deep learning under various reflow conditions to imitate experimental results. Deep learning and machine learning have proven to be useful for physical optimization problems without analytical solutions, particularly for regression and classification tasks. In this application, the self-assembly of cylinder-forming block copolymers (BCPs), confined in prepatterns of the resist reflow process (RFP) to produce small contact hole (C/H) dimensions, was described using the self-consistent field theory (SCFT). This research paves the way for the shrink modeling of the enhanced resist reflow process (RFP) for random contact holes (C/Hs) and the production of smaller contact holes.

## 1. Introduction

Due to the three-dimensional (3D) chip structure, the high cost of extreme ultraviolet (EUV) exposure tools, and the difficulty of multi-patterning, pattern shrinkage processes have generated significant interest in finding effective methods to reduce the feature sizes in microelectronics and data storage devices. These methods are divided into a top-down approach in photolithography [1,2] and a bottom-up approach in self-assembly [3,4]. For technology below the 10-nm scale, top-down exposure tools face obstacles, such as diffraction-limited resolution and the high cost of ownership [5,6,7]. Directed self-assembly, as a bottom-up approach, has problems such as insufficient process support and challenges in mass production [8,9]. However, thermal treatment is a process extension technique that uses current lithography equipment and resists [10]. In lithography, the thermal processes include softbake (SB), post-exposure bake (PEB), and thermal processing after development. These three types of thermal processes are essentially the same in terms of heat treatment but affect critical dimensions (CDs) differently. The purpose of softbake (SB) is to remove excess solvent after spin coating, relieve strain in the solid film, and provide better adhesion to the substrate [11]. Post-exposure bake (PEB) aims to reduce the standing wave effect, thus enhancing the linewidth control and resolution [12,13]. Shrink technologies after development include the resist flow process (RFP) [14], shrink assist film for enhanced resolution (SAFIER^TM^) [15], and resolution enhancement lithography assisted by chemical shrink (RELACS^TM^) [16,17]. Shrink assist film for enhanced resolution (SAFIER™) is a thermal flow with a passive overcoat, which is coated on resist patterns, baked for thermal flow, and then removed by rinsing with water. Shrink assist film for enhanced resolution (SAFIER™) provides mechanical support to contact walls to avoid pattern profile degradation during thermal reflow. Resolution enhancement lithography assisted by chemical shrink (RELACS™) is an active shrink overcoat, which is coated on a patterned wafer and heated (mixed bake) to crosslink and react with the resist film. Resolution enhancement lithography assisted by chemical shrink (RELACS™) exhibits less pitch dependence than the resist flow process (RFP) and shrink assist film for enhanced resolution (SAFIER™) [18]. Finally, the resist flow process (RFP), as a resolution-enhancement technique, is an effective method that does not require additional complex process steps [19,20]. The resist flow process (RFP) reduces the pattern size of a resist by thermally heating it above its glass transition temperature after development. The bonding of the synthesized resist is reduced at temperatures above glass transition, and its mobility is improved. In the case of patterning fine contact holes (C/Hs), which have lower resolution performance compared to line and space patterns due to insufficient depth of focus (DOF) and low aerial-image contrast, the three-dimensional (3D) structure of the synthesized resist is altered. As a result of the additional thermal energy, the contact hole (C/H) patterns shrink. However, predicting the results of the resist flow process (RFP) is challenging because the optical proximity effects (OPEs) of the resist flow process (RFP) become quite severe as critical dimensions (CDs) shrink to sub-10-nm patterns. Many parameters affect the results, such as the baking temperature, the baking time, the original characteristics of the resist, the volume of the resist surrounding contact holes (C/Hs), the initial size and shape of contact holes (C/Hs), and the contact hole (C/H) array. Most empirical models use heuristic methods, such as linear regression, to predict the critical dimensions (CDs) of the reflowed structure. However, these models do not account for intermediate shapes. As a result, no general equations or methods are available for predicting contact hole shrinking.

The objectives of this research were to develop a physically accurate resist model based on an understanding of the mechanistic behaviors that drive photoresist imaging, to achieve the best prediction of resist images across multiple processing conditions and to create a general simulation approach to reduce the optical proximity effects (OPEs) of the resist flow process (RFP). The resist flow process (RFP) is described using the surface evolution method, the finite-element method (FEM), machine learning, and deep learning under various reflow conditions to simulate experimental results. Machine learning and deep learning are particularly useful for physical optimization problems that lack analytical solutions. For applications of the resist flow process (RFP), the self-assembly of cylinder-forming block copolymers (BCPs) confined in prepatterns of the resist flow process (RFP) is described using self-consistent field theory (SCFT).

## 2. Simulation Methods

Figure 1 shows the resist flow process (RFP) in lithography. The resist flow process (RFP) depicted in Figure 1c occurs after the following processes: spin coating (Figure 1a), softbake, exposure, post-exposure bake, and development (Figure 1b). The resist flow process (RFP) reduces the pattern size of the resist by heating it above its glass transition temperature following the development process. At this elevated temperature, the resist’s molecular bonds weaken, enhancing its mobility. This alters the resist’s three-dimensional (3D) structure and results in the shrinkage of the contact hole (C/H) pattern due to the additional thermal energy.

### 2.1. Surface Evolver (SE) Method

The reflow process can be understood as an energy imbalance or surface-tension-driven creep process, where the initial pattern deforms toward a minimal energy shape under constant force due to polymer’s wetting. Surface Evolver is a computer program that minimizes the energy of a surface subject to constraints [21,22,23]. The surface is represented as a simplicial complex. The energy can include surface tension, gravity, and other forms. Constraints may be geometrical constraints on vertex positions or constraints on integrated quantities such as body volumes. Minimization is achieved by evolving the surface along the energy gradient. The condition for a minimum using the gradient descent method and the correcting motion $R_{v}$ of vertex v are
(1)∇f=λ∇g,
(2)$${\vec{R}}_{v}=\sum_{k}c_{k}{\vec{W}}_{vk},\;\sum_{v}{\vec{R}}_{v}{\vec{W}}_{vk}=-\delta_{k},\;\sum_{k}c_{k}\sum_{v}{\vec{W}}_{vk}{\vec{W}}_{v\tilde{k}}=-\delta_{\tilde{k}}$$
where f, g, λ, ${\vec{W}}_{vk}$, and $\delta_{k}$ are a function, a constraint function, a constant known as the Lagrange multiplier, the gradient of the constraint k as a function of the position of vertex v, and the excess value of the quantity k, respectively.

### 2.2. Finite Element Method (FEM)

The finite element method (FEM) is a mathematical (numerical) technique for finding approximate solutions to partial differential equations. For a structure solver, the governing equation of structural analysis by Hooke’s Law, the strain (ϵ)-displacement (u) relation, and the stress (σ)-strain relation are
(3)$$E\frac{\partial^{2}u}{\partial x^{2}}+Q=0,\;\epsilon =\frac{\partial u}{\partial x},\;\sigma =E\epsilon =E\frac{\partial u}{\partial x},$$
where E is the elastic modulus and Q is the axial force per unit length [24,25]. For a thermal conduction solver, which deals with heat transfer through molecular agitation within a material without any motion of the material, the governing equations of heat transfer analysis by Fourier’s law, heat flux (q), thermal strain ($\epsilon_{t}$), and the thermal stress ($\sigma_{t}$)-thermal strain relation are
(4)$$\rho C_{p}\frac{\partial T}{\partial t}-\nabla \cdot \left(k\nabla T\right)=f,\;q=-k\nabla T=-k\frac{\partial T}{\partial n},\;\epsilon_{t}=\alpha \left(∆T\right),\;\sigma_{t}=E\cdot \alpha \left(∆T\right),$$
where T is temperature, k is the thermal conductivity, t is time, ρ is the material density, $C_{p}$ is the specific heat, α is the coefficient of thermal expansion, E is the elastic modulus, and f is the heat generated inside the body [26,27]. The resist reflow above its glass transition temperature can be assumed to be an ideal fluid, which is incompressible and has a constant density, and the force exerted across a geometrical surface element within the fluid. For an incompressible fluid solver, the Navier–Stokes equation as the governing equation and the continuity equation are
(5)$$\frac{\partial \vec{u}}{\partial t}+\left(\vec{u}\cdot \nabla \right)\vec{u}=-\frac{1}{\rho }\nabla p+\frac{1}{\rho }\nabla \cdot \vec{\tau }+f,\;\nabla \cdot \vec{u}=0,$$
where $\vec{u}$, t, $\vec{\tau }$, f, ρ, and p are the flow velocity, time, stress tensor, body force (force per unit mass), density, and pressure, respectively [28,29,30].

### 2.3. Orthogonal Fitting Functions

The resist flow process (RFP) bias $\left(\Delta CD\right)$, which represents the reduction of critical dimensions (CDs) before and after resist reflow, is influenced by the baking temperature $\left(T_{b}\right)$, baking time $\left(t_{b}\right)$, the original characteristics of the resist $\left(K_{r}\right)$, the resist volume surrounding contact hole (C/H) (or the contact hole (C/H) array) $\left(V_{n}\right)$, and the initial contact hole (C/H) size. Under a simple thermal assumption, the actual resist flow process (RFP) bias can be approximated as
(6)$$\Delta CD=f\left(T_{b},t_{b}\right)\cdot f\left(V_{n}\right)\cdot f\left(K_{r}\right),$$
(7)$${\left(\Delta CD\right)}^{2}\approx \left[\alpha_{1}\cdot \exp\left(\alpha_{2}/T\right)\right]\left[\alpha_{3}\cdot \exp\left(\alpha_{4}\cdot \Lambda \;\right)\right],$$ where $\alpha_{1}$, $\alpha_{2}$, $\alpha_{3}$, and $\alpha_{4}$ are constant. The parameter $V_{n}$ (resist volume surrounding contact hole (C/H)) can be replaced by a ratio, Λ, between the contact hole and pitch size. The bias function $f\left(T_{b},t_{b}\right)$ of temperature and time is assumed to be an exponential function due to the relationship between diffusion length and viscosity.

### 2.4. Deep Learning and Machine Learning

For a multivariable linear regression, the linear model functions $f_{w,b}\left(x^{\left(i\right)}\right)$ and the cost functions $J\left(w,b\right)$ with multiple variables are, respectively,
(8)$$f_{w,b}\left(x^{\left(i\right)}\right)=w{\cdot x}^{\left(i\right)}+b,J\left(w,b\right)=\frac{1}{2m}\sum_{i=0}^{m-1}{\left(f_{w,b}\left(x^{\left(i\right)}\right)-y^{\left(i\right)}\right)}^{2},$$
where w and $x^{\left(i\right)}$ are vectors, $y^{\left(i\right)}$ is the target value, $w_{j}$ and b are parameters, and m is the number of features [31]. For classification methods, a logistic regression model applies the sigmoid function to the linear regression model, and the cost functions $J\left(w,b\right)$ are, respectively,
(9)$$f_{w,b}\left(x^{\left(i\right)}\right)=g\left(w{\cdot x}^{\left(i\right)}+b\right)=\frac{1}{1+e^{-\left(w{\cdot x}^{\left(i\right)}+b\right)}},$$
(10)$$J\left(w,b\right)=-\frac{1}{m}\sum_{i=1}^{m}\left[y^{\left(i\right)}log\left(f_{w,b}\left(x^{\left(i\right)}\right)\right)+\left(1-y^{\left(i\right)}\right)log\left({1-f}_{w,b}\left(x^{\left(i\right)}\right)\right)\right]+\;\frac{\lambda }{2m}\sum_{j=1}^{n-1}w_{j}^{2}$$ where λ and w are a regularization constant and a parameter, respectively [32]. For multi-class classification, the softmax function $a_{j}$ of neural networks converts a vector z into a probability distribution:(11)$$a_{j}=\frac{e^{z_{j}}}{\sum_{k=1}^{N}e^{z_{k}}},$$
where $z_{j}=w_{j}\cdot x+b_{j}$ (j=1,⋯,N) and N is number of features (or categories) in the output layer [33,34].

Decision trees (DTs) are a supervised learning approach used in data mining and machine learning [35]. For classification and regression, decision trees (DTs) are simple to understand and interpret by using visualized trees and require little data preparation. Decision trees (DTs) are represented as tree structures, where each internal node represents a feature, each branch represents a decision rule, and each leaf node represents a prediction. The algorithm works by recursively splitting the data into smaller and smaller subsets based on the feature values. At each node, the algorithm chooses the feature that best splits the data into groups with different target values. Gini impurity, as the criterion to measure the impurity or purity of a dataset, is calculated using the following formula:(12)$$Gini\left(t,D\right)=1-\sum_{l\in levels\left(t\right)}{p\left(t=l\right)}^{2},$$
where p represents the proportion of data points belonging to class t in dataset D. Lower Gini impurity values indicate a purer dataset [36].

Support vector machines (SVMs) are a set of supervised learning methods used for classification and regression [37]. A support vector machine (SVM) constructs a hyperplane or set of hyperplanes in a high or infinite dimensional space, which can be used for classification and regression. For classification, the support vector classification (SVC) solves the following primal problem:(13)$$\underset{w,b,\zeta }{\min}\frac{1}{2}w^{T}w+C\sum_{i=1}^{n}\zeta_{i},\;y_{i}\left(w^{T}\phi \left(x_{i}\right)+b\right)\geq 1-\zeta_{i},\;\zeta_{i}\geq 0,\;i=1,\ldots ,\;n$$ where $x_{i}\in R^{p}$ is a training vector, $y\in {\left\{1,-1\right\}}^{n}$ is a vector, C is a regularization parameter, and $\zeta_{i}$ is the distance from their correct margin boundary [38].

### 2.5. Self-Consistent Field Theory (SCFT)

For block copolymer (BCP) self-assembly, self-consistent field theory (SCFT) is used to describe the self-assembly of diblock copolymers confined in prepatterns. Under the mean field approximation, the free energy (F) can be
(14)$$\begin{array}{c} \frac{F}{{nK}_{B}T}=-\ln\left[\frac{Q}{V}\right]-\frac{1}{V}\int d\vec{r}\left[\sum_{k}\omega_{k}\left(\vec{r}\right)\phi_{k}\left(\vec{r}\right)+\sum \chi_{AB}N\phi_{A}\left(\vec{r}\right)\phi_{B}\left(\vec{r}\right)\right.\\ \left.+\xi \left(\vec{r}\right)\left(\sum_{k}\phi_{k}\left(\vec{r}\right)-1\right)\right] \end{array},$$
where V is the dimensionless total volume of the system, $\omega_{k}$ is the mean field of types k (=*A* and *B*), $\chi_{AB}$ is the Flory–Huggins interaction parameter of the two types, $\xi \left(\vec{r}\right)$ is the Lagrange multiplier enforcing the incompressibility constraint, N is the polymerization degree, and *Q* $\left(=\left(1/V\right)\int d\vec{r}q\left(\vec{r},1\right)\right)$ is a single-molecule partition function in mean fields [39,40,41,42]. Diffusion dynamics can model the time evolution of the system:(15)$$\frac{\partial }{\partial t}\phi_{K}\left(\vec{r},t\right)=L\nabla^{2}\mu_{K}\left({\vec{r}}^{\prime },t\right)+\eta_{K},$$
where $\phi_{K}$ is the block concentration of type *K* (=*A* or *B*), L is the constant mobility, $\mu_{K}$ is the chemical potential, and $\eta_{K}$ is the thermal noise [43,44,45]. By minimizing free energy for densities and mean fields,
(16)$$\phi_{A}\left(\vec{r}\right)+\phi_{B}\left(\vec{r}\right)=1,\;\omega_{K}\left(\vec{r}\right)=\chi_{AB}N\phi_{K}\left(\vec{r}\right)-H\left(\vec{r}\right)+\xi \left(\vec{r}\right),$$
where χ is the Flory–Huggins interaction parameter, N is the polymerization degree, H is the polymer–surface interaction, and ξ is the Lagrange multiplier enforcing the incompressibility constraint. The monomer density, due to the partial partition function, is
(17)$$\phi_{K}\left(\vec{r}\right)=\frac{V}{Z}\int_{a}^{b}q_{A}\left(\vec{r},s\right)q_{B}\left(\vec{r},s\right)ds,$$
where V is the dimensionless total volume of the system, Z is the single-chain partition function in the independent particle approximation imposed by mean field theory, $q_{K}\left(\vec{r},s\right)$ is the diffusion equation of the partition function due to the contour length (s) of a copolymer chain, and the integral intervals (a,b) are (0,f) (or (f,1)) at K=A(or B)

## 3. Experiment

For a 193-nm argon fluoride (ArF) chemically amplified resist (CAR), a 6% transmittance attenuated phase shift mask (PSM), quaterpole off-axis illumination (OAI), argon fluoride (ArF) 193-nm illumination, and a 0.75 numerical aperture (NA) were used. The chemically amplified resist (CAR) thickness was 0.35 μm with a bottom antireflection coating (BARC) of 0.082 μm. For thermal reflow, an antireflective layer of 80-nm thick resist was coated over the silicon wafer prior to the resist process. The coated thickness was 0.37 μm. An ethylvinylether-based polymer was coated and prebaked at 100 ℃ for 60 s. The exposure system was an ASML-700 with a numerical aperture (NA) of 0.6, a partial coherency (σ) of 0.4, and an attenuated phase-shift mask. Exposed wafers were baked at 110 ℃ for 60 s on a hot plate and developed in 2.38 wt% tetramethyl ammonium hydroxide (TMAH) aqueous solution for 60 s. The resist reflow time was 90 s. Figure 2 shows the experimental results from Hynix Semiconductor Inc. for various temperature and duty ratios. The contact holes (C/Hs) become smaller as the duty ratio and temperature increased [46].

## 4. Results and Discussion

### 4.1. Surface Evolver (SE)

Figure 3 shows the simulation outcomes generated using Surface Evolver (SE), a computer program by Kenneth Brake, using Equations (1) and (2). Surface Evolver (SE) treats three-dimensional (3D) structures as surfaces governed purely by energy optimization criteria. After the resist flow process (RFP), the surface energy and surface area of one contact hole and nine contact holes were reduced due to simulation runtimes, as shown in Figure 3a. Figure 3c,d display the reflowed images of one contact hole versus the simulation runtimes for the surface areas of 38.3685 and 32.9430, respectively. Figure 3f,g illustrate the reflowed images of nine contact holes for the surface areas of 155.0468 and 144.5697, respectively. However, the Surface Evolver (SE)-based model did not require explicit knowledge of material-specific parameters and predicted the dependence of shape evolution on the simulation runtime.

### 4.2. The Results of the Finite Element Method (FEM)

Figure 4 shows the simulation results generated through MATLAB to analyze the resist reflow through thermal cycling at hardbake. A mesh with elements no larger than 0.1 was generated, as illustrated in Figure 4a. The temperature distribution in Figure 4b was calculated from the steady-state solution of Equation (4). For both constant thermal conductivity and temperature-dependent thermal conductivity, Figure 4c illustrates the temperature–time relationship in typical three-stage wafer proximity contact processing, including heating by the preheated hotplate, transfer by the wafer carrier from the hotplate to the chillplate, and cooling by the chillplate. The simulation conditions were mass density = 1, specific heat of the material = 1, constant thermal conductivity of the material = 1, and temperature-dependent thermal conductivity = 0.003 × temperature. Compared with the constant-conductivity case, the temperature on the right-hand edge is lower, as shown in Figure 4c. This is due to the lower conductivity in regions with lower temperatures.

Figure 5 shows the contact hole (C/H) deformation obtained using the finite element method (FEM) in the static structural ANSYS program (Student Edition 2024) to analyze heating at the preheated hotplate for one contact hole (C/H) shape. Figure 5a presents a structure consisting of a 5 mm×5 mm×5.6 mm rectangular resist, a 5 mm×5 mm×0.8 mm rectangular silicon substrate, and an emptied cylinder with a height of 5 mm and a radius of 1.05 mm. The ANSYS model consisted of 83,878 nodes and 18,421 elements for a 200 μm mesh size, as shown in Figure 5b. Figure 5b,d show the three-dimensional (3D) plots of contact hole (C/H) thermal deformation at 0 s and 20 ℃, 0.5 s and 74 ℃, and 1 s and 168 ℃. The maximum thermal deformation was 41.643 μm at 1 s and 168 ℃, as shown in Figure 5c. The simulation conditions for a silicon substrate were density $\rho =2330\;kg/m^{3}$, Young’s modulus E=150 GPa, Poisson’s ratio v=0.28, and the coefficient of thermal expansion was α=0.9 ppm/K; the simulation conditions for the resist were density $\rho =950\;kg/m^{3}$, Young’s modulus E=2 GPa, Poisson’s ratio v=0.33, and the coefficient of thermal expansion was α=60 ppm/K. However, a limitation of ANSYS based on the finite element method (FEM) arises from meshing thicknesses greater than 10 μm and shear loading, which can cause an error in the finite element method (FEM) due to the impact on the stiffness matrix from variations in the length-to-thickness ratio of the beam element [47].

Figure 6 shows a phase change of a contact hole (C/H) sidewall due to a glass transition temperature. The melting boundaries, influenced by gravity, moved from the left side to the right side at 0, 1, 30, 50, 100, 150, and 200 simulation steps. The boundary behavior illustrates why the top contact hole (C/H) diameter was more variable than the bottom contact hole (C/H) diameter after the resist reflow. The simulation conditions were a simulation area of 8.89 mm × 6.35 mm; 3721 nodes and 3600 elements for a 0.546 mm mesh size after meshing; material properties (density of $750\;kg/m^{3}$, specific heat of 1800 J/(kg·K), thermal conductivity of 2 W/(m·K), viscosity of 0.00181 kg/(m·s), thermal expansion coefficient of $0.00012\;K^{-1}$, pure solvent melting heat of 174,000 J/kg, solidus temperature of 294 K, and liquidus temperature of 297 K); boundary conditions (left wall at 363 K and right wall at 294 K); solution controls under relaxation factors (pressure of 0.3, density of 1, body forces of 1, momentum of 0.8, turbulent kinetic energy of 0.8, specific dissipation rate of 0.8, turbulent viscosity of 1, liquid fraction update of 0.2, and energy of 0.7); and initial values (gauge pressure of 0 Pa, X-axis velocity of 0 m/s, Y-axis velocity of 0 m/s, turbulent kinetic energy of 1 $m^{2}/s^{2}$, specific dissipation rate of 1 $s^{-1}$, and temperature of 294 K).

### 4.3. Results of Orthogonal Fitting Functions and Deep Learning

Figure 7 shows the comparison between the experiment results and the simulation results of thermal reflow biases dependent on temperature and pitch sizes in a 193-nm chemically amplified resist (CAR). Depending on the aspect ratio, which is the ratio between the contact hole size and the pitch size, the simulated results from an orthogonal fitting function, linear regressions using a Python program (version 3.12.2) and Scikit-Learn, and a convolutional neural network (CNN) exhibited mismatches with the experimental results within different error ranges. Although the simulation parameters do not account for the chemical phenomenon of thermal reflow, the critical dimension bias after resist reflow can be predicted in a linear system using the fitting function of the experimental data. From Equation (7), the simulated function of thermal bias for a 193-nm chemically amplified resist (CAR), using an orthogonal fitting functional method, was
(18)$${\left(\Delta CD\right)}_{Thermal\;reflow}^{2}=\left[\alpha_{1}\times \exp\left(\alpha_{2}/T\right)\right]\left[\alpha_{3}\times \exp\left(\alpha_{4}\times \Lambda \;\right)\right]\;=\left[3.696\times 10^{29}\times \exp\left(-8347.64393/T\right)\right]\left[1.88152\times \exp\left(-1.90217\times \Lambda \right)\right]$$ where T is temperature and Λ is the ratio between the contact hole size and the pitch size, as shown in Figure 7a. Figure 7b,c show comparisons of linear regressions from a Python program and Scikit-Learn to the experimental results using square root scaling. For the Python program, after 200,000 iterations, the fitting function was $y=0.69x_{1}+0.81x_{2}-107.739906$, in which y is ${log}_{10}(thermal\;reflow\;bias)$, $x_{1}$ is aspect ratio, and $x_{2}$ is temperature, as shown in Figure 7b. For Scikit-Learn, the fitting function was $y=0.69x_{1}+0.81x_{2}-107.739888$, as shown in Figure 7c. For the convolutional neural network (CNN) regression, the network architecture comprised of four layers (Conv1D(finters = 64), Dense(64), Dense(16), and Dense(1)). The results of the convolutional neural network (CNN) regression were better than those of the fitting function and linear regressions, as shown in Figure 7d.

Figure 8 shows the classifications of a logistic regression in deep learning using TensorFlow (version 2.17.0) and the decision surfaces of a decision tree using Scikit-Learn for binary and multiple classes. For binary classification, logistic regression with a sigmoid function in TensorFlow, after 100,000 epochs, resulted in a loss value of 0.4074 and a classified linear equation of $y=-\left(1/0.55\right)\times 1.22x-79.41$ with one dense layer and three parameters, as shown in Figure 8a. For the multiple classification with logistic regression, after 250 epochs, the loss value was 1.0506, and the classified linear equations were $y=-\left(1/0.55\right)\cdot 1.22x-79.41$ and $y=-\left(1/0.55\right)\times 1.21x-81.4$ with two dense layers and 15 parameters, as shown in Figure 8b. For the decision surfaces of a decision tree using Scikit-Learn (version 1.3.2), the accuracies for the binary class with a max depth of two and the multiple class with a max depth of three were 0.875 and 0.7916, respectively, as shown in Figure 8b,d. Figure 8e shows a decision tree for multiple classes with a max depth of four.

Figure 9 shows how to plot the decision surface for four support vector machine (SVM) classifiers with different kernels, which are a support vector classification (SVC) with a linear kernel, LinearSVC (linear kernel), a support vector classification (SVC) with a radial basis function (RBF) kernel, and a support vector classification (SVC) with a polynomial (degree 3) kernel. The linear models, such as support vector classification (SVC) (kernel = “linear”) and LinearSVC(), yield insufficient decision boundaries for non-linear decision data, as shown in Figure 9a,b. The non-linear kernel models (Gaussian RBF and polynomial) provide more flexible non-linear decision boundaries, as shown in Figure 9c,d.

Figure 10 shows a comparison between the predicted pattern types from a convolutional neural network (CNN) and the experimental pattern types based on the side images of reflow resist. The network architecture comprised a dense layer with 25 units and a ReLU activation function, a second dense layer with 15 units and a ReLU activation function, and a third dense layer with 10 units and a linear activation function, totaling 1,638,975 parameters. A softmax function was used to calculate the probabilities. After 100 epochs, the loss value was $5.5164\times 10^{-5}$, as shown in Figure 10. The simulated results agree well with the experimental results [48].

### 4.4. Results of Self-Consistent Field Theory (SCFT)

Figure 11 illustrates the shrinking process of cylinder-forming block copolymers (BCPs). The PS-b-PMMA block copolymers (BCPs) are spin-coated into contact holes that have shrunk after thermal reflow. After thermal annealing and wet development in block copolymer (BCP) lithography processes [49], PMMA is removed from the contact hole. The main benefits of this approach are smaller critical dimensions, improved contact hole uniformity, and reduced contact edge roughness [50]. Block copolymer (BCP) lithography can produce smaller contact hole (C/H) dimensions than other lithographic methods, as well as better contact hole uniformity and reduced contact edge roughness.

Figure 12 shows a comparison of the simulation results using rectangular guiding patterns $\left({CD}_{guiding}\right)$ to the experimental results using cylindrical guiding patterns $\left({CD}_{guiding}\right)$ from Reference 49 for the morphologies obtained after the directed self-assembly (DSA) of cylindrical PS-b-PMMA block copolymers (BCPs). The simulation results from self-consistent field theory (SCFT) were similar to the experimental results from Reference 49 for direct self-assembly (DSA) contact hole (C/H) shapes with the prepattern $\left({CD}_{guiding}\right)$ and a BCP natural period $\left(L_{0}\right)$. Simulation conditions included PMMA(7)PS(20) with χN=27, PMMA(10)PS(27) with χN=35, and PMMA(12) PS(34) with χN=46. As one of simulation parameters, the numbers in brackets represent the lengths of the components. Specifically, the numbers of PMMA and PS correspond to their lengths as a proportion of the total number (natural period ($L_{0}$) of the block copolymer (BCP)). These lengths are related to the molecular weights of the copolymers.

## 5. Conclusions

For contact hole shrinkage by resist reflow, the Surface Evolver method, which is a gradient descent technique, reduces the resist volume by decreasing the surface energy of the resist. A disadvantage of this method is the requirement to know the initial surface energy value. The finite element method (FEM) can be easily integrated with commercial tools such as ANSYS and SIEMENS. However, it necessitates the simultaneous implementation of a structural analysis, thermal analysis, and fluid mechanical analysis, for which the resist characteristics in each case must be known. Deep learning does not require knowledge of the resist characteristics. However, the error range should be below 5% for practical applications, necessitating more training and advanced algorithms. Although adding the block copolymer process increases the process complexity, using block copolymers after contact hole shrinkage by resist reflow can improve contact hole uniformity and reduced line edge roughness (LER). Thus, block copolymers can further reduce contact hole patterns. These models can aid in the shrinkage modeling of random contact holes and the fabrication of smaller contact holes.

## Figures and Tables

**Figure 1 micromachines-15-01151-f001:**
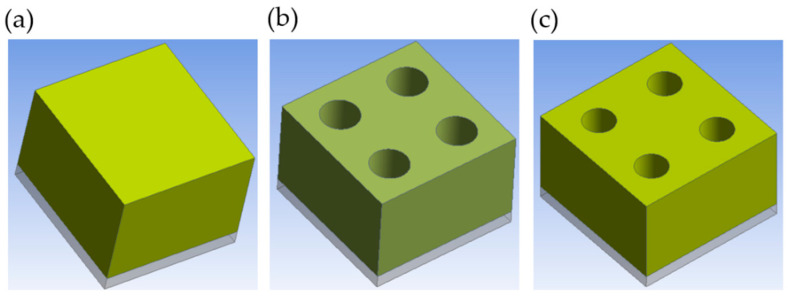
Schematic representation of the resist flow process (RFP): (**a**) spin coating; (**b**) contact hole (C/H) patterns after development process; and (**c**) contact hole (C/H) patterns after thermal reflow. The structure consists of a silicon substrate and resist.

**Figure 2 micromachines-15-01151-f002:**
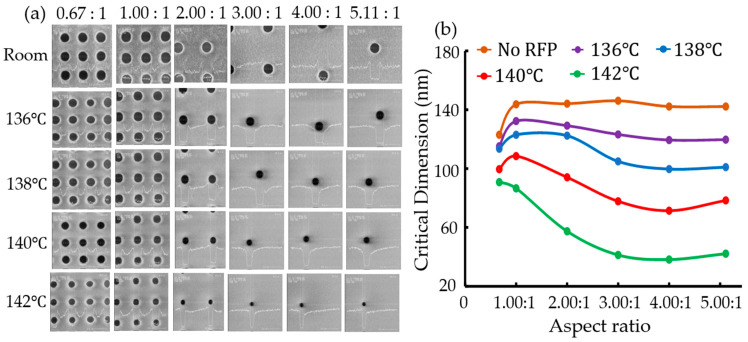
(**a**) Scanning electron microscope (SEM) images of contact holes (C/Hs) and (**b**) a graph of critical dimension (CD) with a function of temperature and duty ratio after the thermal resist process. The upper labels of the images in Figure 2a represent the aspect ratio, which is the ratio between the contact hole size and the pitch size.

**Figure 3 micromachines-15-01151-f003:**
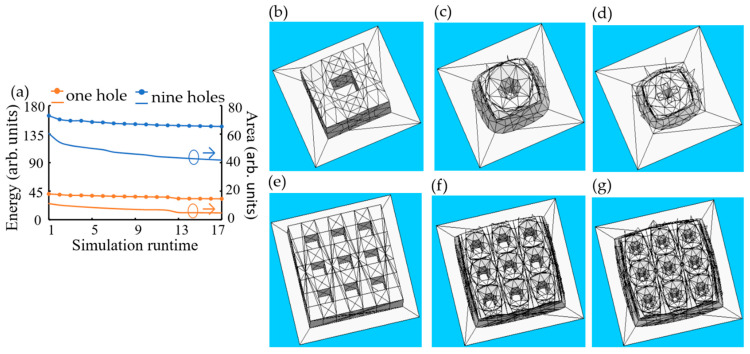
Simulation results of Surface Evolver: (**a**) surface energy and surface area for one contact hole pattern and nine contact holes in terms of simulation times, one contact hole and nine contact holes (**b**,**e**) before the resist flow process (RFP), at surface areas of (**c**) 38.3685, (**d**) 32.9430, (**f**) 155.0468, and (**g**) 144.5697 after the resist flow process (RFP), respectively.

**Figure 4 micromachines-15-01151-f004:**
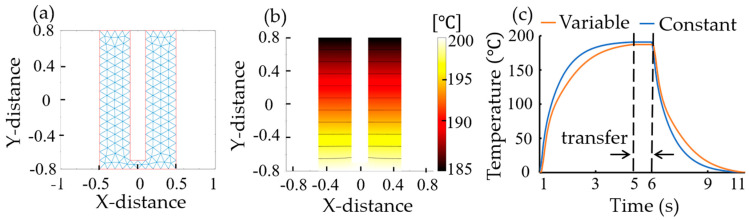
Heat transfer analysis generated through MATLAB: (**a**) a block with the finite element mash displayed, (**b**) a temperature contour with constant thermal conductivity, and (**c**) a plot of simulated bake cycle profiles with constant thermal conductivity (or constant) and temperature-dependent thermal conductivity (or variable), respectively. Transfer time is 1 s.

**Figure 5 micromachines-15-01151-f005:**
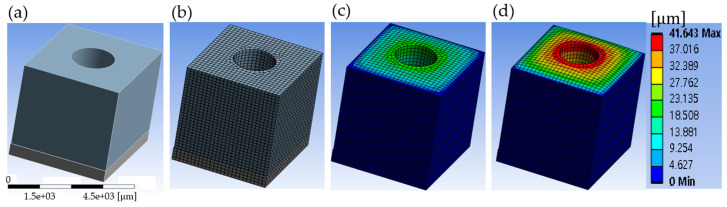
Static structural deformation of one contact hole (C/H) due to heat transfer, generated through ANSYS: (**a**) the structure of a 5 mm×5 mm×0.8 mm bottomplate, a 5 mm×5 mm×5.6 mm upper plate, and an emptied cylinder with a height of 5 mm and a radius of 1.05 mm, and plots of (**b**) mashing and deformation at (**c**) 0.5 s and (**d**) 1 s simulation times.

**Figure 6 micromachines-15-01151-f006:**
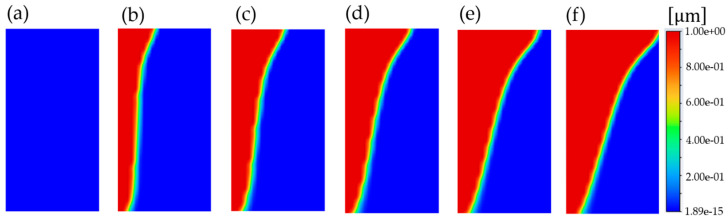
Phase change of a contact hole (C/H) sidewall due to gravity from solid to liquid during thermal reflow, generated using the ANSYS Fluent (Student Edition 2024): melting boundaries at (**a**) 0 steps, (**b**) 1 step, (**c**) 30 steps, (**d**) 50 steps, (**e**) 100 steps, and (**f**) 150 steps.

**Figure 7 micromachines-15-01151-f007:**
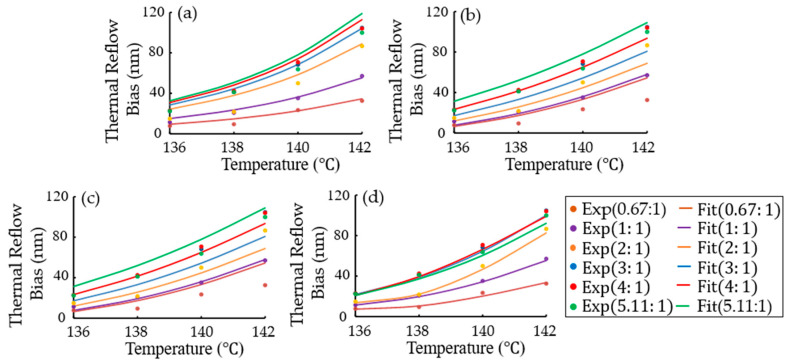
Experimental and simulation results for thermal reflow biases depending on temperatures and duty ratios in a 193-nm argon fluoride (ArF) chemically amplified resist (CAR): the results of (**a**) an orthogonal fitting function, linear regressions from (**b**) a Python program and (**c**) Scikit-Learn, and (**d**) a convolutional neural network (CNN). “Exp” and “Fit” represent the experimental results and simulation results, respectively.

**Figure 8 micromachines-15-01151-f008:**
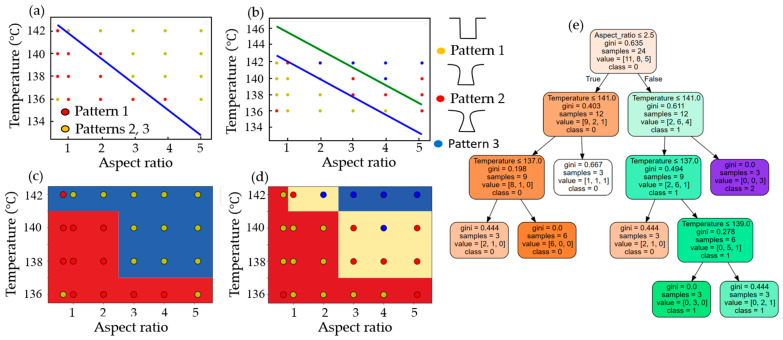
Classifications from a logistic regression in deep learning and a decision tree in machine learning: (**a**,**b**) are binary and multiple classifications from TensorFlow, respectively, (**c**,**d**) are the decision surfaces of a decision tree for the binary class with the max depth of two and the multiple class with a max depth of three, respectively, and (**e**) a decision tree for multiple classes with a max depth of four.

**Figure 9 micromachines-15-01151-f009:**
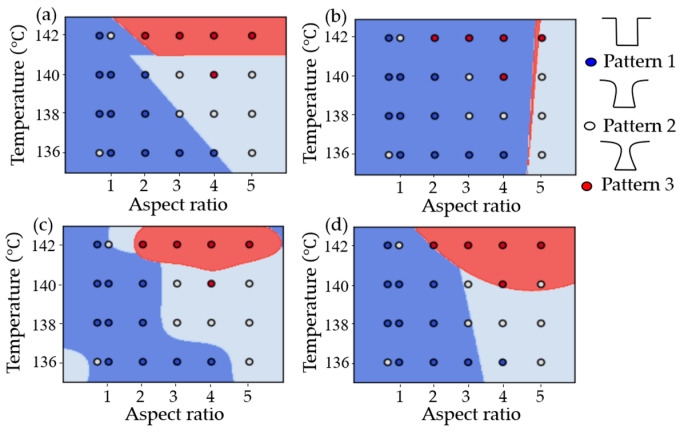
Multi-class classification using a support vector machine (SVM): (**a**) support vector classification (SVC) with a linear kernel, (**b**) LinearSVC (linear kernel), (**c**) support vector classification (SVC) with a radial basis function (RBF) kernel, and (**d**) support vector classification (SVC) with a polynomial (degree 3) kernel.

**Figure 10 micromachines-15-01151-f010:**
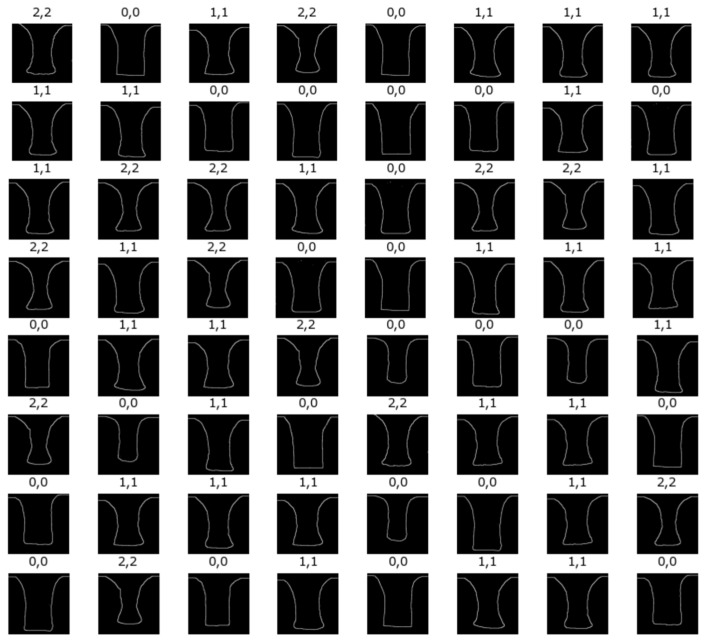
Classification of resist flow process (RFP) side images using a convolutional neural network (CNN). For the upper labels of the images, the first number represents the pattern type from experimental results, and the second number represents the predicted pattern type from a convolutional neural network (CNN).

**Figure 11 micromachines-15-01151-f011:**
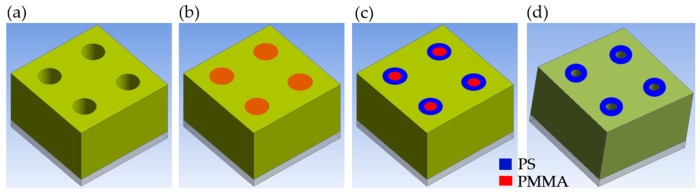
Schematic representation of the self-assembly of cylinder-forming block copolymers (BCPs) confined in cylindrical prepatterns: (**a**) contact hole (C/H) shrinkage patterns after thermal reflow (baking), (**b**) spin coating for PS-b-PMMA block copolymers (BCPs), (**c**) thermal annealing, and (**d**) wet development.

**Figure 12 micromachines-15-01151-f012:**
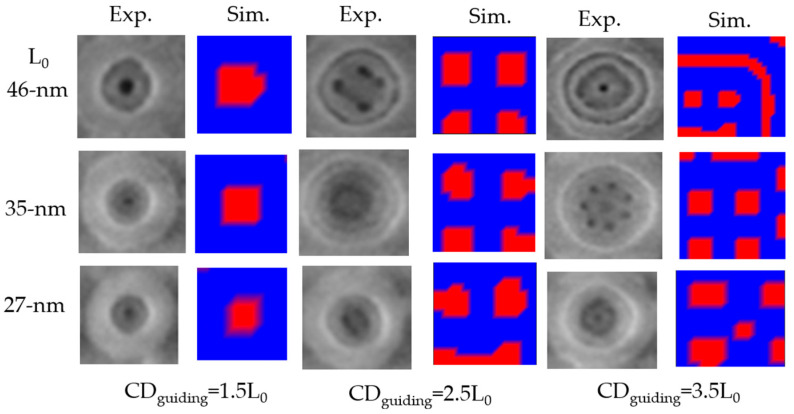
Comparison of the simulation results (Sim.) using rectangular guiding patterns $\left({CD}_{guiding}\right)$ to the experimental results (Exp.) using cylindrical guiding patterns $\left({CD}_{guiding}\right)$ from Reference [49] for the resulting morphologies obtained after the directed self-assembly (DSA) of cylindrical PS-b-PMMA block copolymers (BCPs). $L_{0}$ is a natural period of the block copolymer (BCP), which is directly correlated to the molecular weight of the block copolymer (BCP).

## Data Availability

The data, if required for reproduction, are available to obtain from the corresponding author.

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
