# Peer review of "Contact Hole Shrinkage: Simulation Study of Resist Flow Process and Its Application to Block Copolymers"

_micromachines, 2024, doi:10.3390/mi15091151_

Round 1

Reviewer 1 Report

Comments and Suggestions for Authors

This paper studies the reduction of contact hole sizes in 3D structure chips. Multiple research methods are applied.

Some comments:

1.For the finite element method, a purely linear elastic model may be insufficient and inaccurate for polymers.

2.The results of the FEA model should be validated.

3.In section 2.4 on deep learning and machine learning methods, the author discusses linear regression, decision trees, and support vector machines, which are basic statistical learning methods. Even the convolutional neural network (CNN) mentioned is not considered "deep."

Author Response

Response for a Reviewer

Reviewer #1: This paper studies the reduction of contact hole sizes in 3D structure chips. Multiple research methods are applied.

Some comments:

1. For the finite element method, a purely linear elastic model may be insufficient and inaccurate for polymers.

(Answer) Thank you. You are correct that ANSYS cannot be applied to model resist reflow due to its use of a linear elastic model. The mechanical and thermal behaviors of polymer resist are indeed governed by a non-linear elastic model. Furthermore, resist properties such as thermal conductivity (k), specific heat (Cp), coefficient of thermal expansion (α), and elastic modulus (E) are temperature-dependent, not constant.

However, this paper employed ANSYS with the finite element method to model resist reflow. The author aimed to understand the differences (or error range) in shrinkage sizes predicted by the linear elastic model compared to experimental data. When considering only experimental data before and after resist reflow, ANSYS can be applied within the linear region where shrinkage size varies linearly with temperature. Unfortunately, author could not integrate mechanical, thermal, and fluid analysis for resist reflow using ANSYS, leading to incomplete results.

2. The results of the FEA model should be validated.

(Answer) Thank you. Author was unable to integrate mechanical, thermal, and fluid analysis for resist reflow using ANSYS based on the finite element method. As a result, a comparison between the experimental data and ANSYS results was not performed in the paper. This topic remains for future work.

3. In section 2.4 on deep learning and machine learning methods, the author discusses linear regression, decision trees, and support vector machines, which are basic statistical learning methods. Even the convolutional neural network (CNN) mentioned is not considered "deep."

(Answer) Thank you. Generally, when searching for “CNN” on Google, most websites refer to “deep learning CNN.” Author wishes to retain the term "deep" in this paper.

Reviewer 2 Report

Comments and Suggestions for Authors

The article by Kim S.-K. describes various mathematical modeling techniques for predicting the size and shape of shrinking contact holes over time during the fabrication of three-dimensional structure chips. The author uses surface evolver method, finite element method, self-consistent field theory, orthogonal fitting functions, and deep learning and gives the results of these methods with a brief description of their advantages and disadvantages. The article is concise, well written, and can be published after some improvements.

Specific comments are as follows.

Line 8: “where CD=k1λ/NA”. A brief definition for λ needs to be given.

Lines 127, 150, 162: “the actual RFP bias can be approximated as”, “Decision Trees (DTs) are a supervised learning approach used in data mining and machine learning”, “Support vector machines (SVMs) are a set of supervised learning methods used for classification and regression”. Supporting references would be valuable.

Figure 2, caption. It is necessary to write what the horizontal labels at the top of SEM images mean. In addition, it is also necessary to describe the composition and characteristics of the copolymer and to clarify the definition of the aspect ratio (i.e., it is unclear the aspect ratio between what elements the author mean).

Line 201: “ethylvinylether-based polymer”. The full chemical name of the copolymer should be provided, giving also its block structure, co-monomer mass fractions, and total molecular weight.

Line 350: “After exposure and development, PMMA is removed from the contact hole”. This point is unclear. How could PMMM disappear while PS remained? PMMA and PS are chemically bonded as a block copolymer. A clarification should be added.

Line 366: “PMMA(7)PS(20)”. The meaning of the numbers in brackets is unclear. It would be desirable to specify the mole or weight fractions of co-monomers and the molecular weights of the copolymers.

Line 378: “the block copolymer process adds complexity”. It would be valuable to overview possible block copolymers and their characteristics (co-monomer composition, molecular weight, di-, tri- or multiblock structure, morphology of microdomains during microphase separation) in the article (not in the conclusion).

The author uses many abbreviations, which makes the text poorly perceived. The author could either reduce the use of abbreviations or list them at the beginning or the end of the article.

Author Response

Reviewer #1: The article by Kim S.-K. describes various mathematical modeling techniques for predicting the size and shape of shrinking contact holes over time during the fabrication of three-dimensional structure chips. The author uses surface evolver method, finite element method, self-consistent field theory, orthogonal fitting functions, and deep learning and gives the results of these methods with a brief description of their advantages and disadvantages. The article is concise, well written, and can be published after some improvements.

Specific comments are as follows.

1. Line 8: “where CD=k1λ/NA”. A brief definition for λ needs to be given.

(Answer) Thank you. λ is defined in paper.

In line 8, in another paper

λ is wavelength,

2. Lines 127, 150, 162: “the actual RFP bias can be approximated as”, “Decision Trees (DTs) are a supervised learning approach used in data mining and machine learning”, “Support vector machines (SVMs) are a set of supervised learning methods used for classification and regression”. Supporting references would be valuable.

(Answer) Thank you. In line 127, Equations 6 and 7 are author’s own fitting functions, so, there are no references. In lines 157 and 162, the relevant references were in lines 161 and 169, respectively. For clarity, one of these references can be moved to the corresponding sentences.

In line 163, in another paper

Decision Trees (DTs) are a supervised learning approach used in data mining and machine learning [35].

In line 175, in another paper

Support vector machines (SVMs) are a set of supervised learning methods used for classification and regression [37].

3. Figure 2, caption. It is necessary to write what the horizontal labels at the top of SEM images mean. In addition, it is also necessary to describe the composition and characteristics of the copolymer and to clarify the definition of the aspect ratio (i.e., it is unclear the aspect ratio between what elements the author mean).

(Answer) Thank you. The aspect ratio refers to the ratio between the contact hole size and pitch size. The composition and characteristics of the resist copolymer were not disclosed, as they are proprietary to Hynix Semiconductor Inc.  

In line 208, in another paper

The upper labels of the images in Figure 2(a) represent the aspect ratio, which is the ratio between the contact hole size and the pitch size.

4. Line 201: “ethylvinylether-based polymer”. The full chemical name of the copolymer should be provided, giving also its block structure, co-monomer mass fractions, and total molecular weight.

(Answer) Thank you. The composition and characteristics of the resist copolymer are not due to the proprietary information of Hynix Semiconductor Inc.

5. Line 350: “After exposure and development, PMMA is removed from the contact hole”. This point is unclear. How could PMMM disappear while PS remained? PMMA and PS are chemically bonded as a block copolymer. A clarification should be added.

(Answer) Thank you. In the paper, exposure and development are modified to thermal annealing and wet development. Also, reference is attached in the sentence. 

In line 371, in another paper

(c) thermal annealing, and (d) wet development.

In line 375, in another paper

After thermal annealing and wet development in block copolymer (BCP) lithography processes [49],

6. Line 366: “PMMA(7)PS(20)”. The meaning of the numbers in brackets is unclear. It would be desirable to specify the mole or weight fractions of co-monomers and the molecular weights of the copolymers.

(Answer) Thank you. The numbers in brackets represent the lengths of the component used as simulation parameters. Specifically, the numbers of PMMA and PS correspond to their lengths as a proportion of the total number (the block copolymer (BCP) natural period ). These lengths are related to the molecular weights of the copolymers.      

In line 394, in another paper

The numbers in brackets represent the lengths of the component used as simulation parameters. Specifically, the numbers of PMMA and PS correspond to their lengths as a proportion of the total number (the block copolymer (BCP) natural period ()). These lengths are related to the molecular weights of the copolymers.     

7. Line 378: “the block copolymer process adds complexity”. It would be valuable to overview possible block copolymers and their characteristics (co-monomer composition, molecular weight, di-, tri- or multiblock structure, morphology of microdomains during microphase separation) in the article (not in the conclusion).

(Answer) Thank you. The sentence stating that the block copolymer process adds complexity has been revised to indicate that adding the block copolymer process increases process complexity.

In line 401, in another paper

Although adding the block copolymer process increases process complexity,

8. The author uses many abbreviations, which makes the text poorly perceived. The author could either reduce the use of abbreviations or list them at the beginning or the end of the article.

(Answer) All abbreviations in the paper have been expanded to their full forms.
